# N-cadherin mechanosensing in ovarian follicles controls oocyte maturation and ovulation

Alaknanda Emery[1], Orest W Blaschuk[2], Doan T Dinh[1], Tim McPhee[1], Rouven Becker[3], Andrew D Abell[3], Krzysztof M Mrozik[4], Andrew CW Zannettino[5], Rebecca L Robker[1], Darryl L Russell[1]*

[1]Robinson Research Institute, Adelaide Medical School, The University of Adelaide, Adelaide, Australia; [2]Zonula Incorporated, Kirkland, Canada; [3]Department of Chemistry, The University of Adelaide, Adelaide, Australia; [4]Myeloma Research Laboratory, Adelaide Medical School, Faculty of Health and Medical Sciences, The University of Adelaide, Adelaide, Australia. South Australian Health and Medical Research Institute, Adelaide, Australia; [5]Myeloma Research Laboratory, Adelaide Medical School, Faculty of Health and Medical Sciences, The University of Adelaide, Adelaide, Australia. South Australian Health and Medical Research Institute, Adelaide, Australia. Central Adelaide Local Health Network, Adelaide, Australia

*For correspondence:
darryl.russell@adelaide.edu.au

## eLife assessment

The manuscript describes **important** findings regarding the significance of CHD2 in ovarian folliculogenesis. Overall, the results lead to **convincing** conclusions, with minimal concerns raised by the reviewers. Both the results and conclusions are well discussed. This work will be of interest to ovarian biologists and physicians working on female fertility.

**Abstract** The cell adhesion molecule N-cadherin (CDH2) is a membrane component of adherens junctions which regulates tissue morphogenesis and architecture. In the follicles of mammalian ovaries, N-cadherin adherens junctions are present between granulosa cells, cumulus cells, and at the interface of cumulus cell transzonal projections and the oocyte. We demonstrate a mechanosensory role of N-cadherin integrating tissue structure and hormonal regulation of follicular morphogenic events including expansion of the cumulus–oocyte complex (COC) matrix, oocyte maturation, and ovulation. Two small molecule N-cadherin antagonists inhibited COC maturation in vitro. Transcriptome profiling revealed that targets of β-catenin and YAP1 pathways were dysregulated by N-cadherin antagonists. In vivo, N-cadherin antagonist significantly reduced ovulation in mice compared to controls (11 vs 26 oocytes/ovary; $p = 5.8 \times 10^{-6}$). Ovarian follicles exhibited structural dysgenesis with granulosa and cumulus cell layers becoming disorganised and the connection between cumulus cells and the oocyte disrupted and the transcriptome again indicated altered mechanical sensing causing dysregulation of the Hippo/YAP and β-catenin pathways and extracellular matrix reorganisation. Granulosa-specific N-cadherin depletion in $Cdh2^{Fl/FL};Amhr2^{Cre/+}$ also showed significantly altered mechanosensitive gene expression and reduced ovulation. Our findings demonstrate a critical role for N-cadherin in ovarian follicular development and ovulation, and the potential to inhibit ovulation through targeting this signalling mechanism.

## Introduction

Classical (type I) cadherins are calcium-dependent cell adhesion molecules that form intercellular adherens junctions thereby regulating cell–cell recognition and the assembly of tissues. These cadherins control a myriad of biological processes including cell migration, proliferation, death, and adhesion (*Pannekoek et al., 2019*; *Mrozik et al., 2018*). They consists of extracellular, transmembrane, and intracellular domains. The intracellular domain interacts with a complex composed of β-catenin, α-catenin, p120, and α-actinin that links the type I cadherins to the actin cytoskeleton (*Kourtidis et al., 2017*; *Mrozik et al., 2020*). The formation of adherens junctions requires interaction of cadherins within the same membrane to form lateral or *cis*-dimers. These *cis*-dimers then promote intercellular adhesive dimerisation or *trans*-dimers of cadherins on adjacent cells (*Vendome et al., 2014*) to form adherens junctions. Intracellular signalling pathways are activated by mechanical force generated through cell–cell and cell–matrix interactions. Two types of cell surface receptors, cadherins and integrins directly mediate cell adhesion and activate intracellular pathways such as β-catenin (*Fernández-Sánchez et al., 2015*). Cross-talk between the classical cadherin, N-cadherin (CDH2), and the Hippo signalling pathway is another mechanosensitive signalling axis that regulates cell proliferation and determines organ size (*Benham-Pyle et al., 2015*). The Hippo-YAP1 signalling is regulated by numerous upstream signals including organ size, extracellular matrix (ECM) stiffness, and cell polarity (*Vite et al., 2018*).

Ovarian follicles are complex multilayered structures required for the development and maturation of oocytes. Follicles also serve as an endocrine organ coordinating the female reproductive cycle, augmenting oocyte maturation and ovulation (the release of the oocyte) at the appropriate time for fertilisation. In growing follicles, the cumulus cells surrounding the oocyte provide essential metabolites and maintain oocyte meiotic arrest through transfer of cAMP and cGMP from cumulus cells to oocyte. This communication requires thin cytoplasmic projections known as transzonal projections (TZPs) that extend from the cumulus cells, through the thick ECM of the zona pellucida to contact the oocyte plasma membrane (*Baena and Terasaki, 2019*). Gap junctions at the cumulus–oocyte cellular contacts transport small molecules such as glucose metabolites, ATP, and cAMP between the cells (*Fontana et al., 2020*). Adherens junctions have been proposed to stabilise these cumulus cell–oocyte contacts as a necessary precursor to gap junction formation (*El-Hayek et al., 2018*), and the contacts are selectively uncoupled just prior to ovulation (*Abbassi et al., 2021*).

Contacts between cumulus and granulosa cell layers are critical for the progressive development of ovarian follicles and maturation of oocytes (*Baena and Terasaki, 2019*; *Prunskaite-Hyyryläinen et al., 2014*; *Rowlands et al., 2000*). N-cadherin is present on the intercellular junctions between granulosa cells (*Jiang et al., 2017*) and β-catenin has been observed on granulosa and cumulus cell membranes, while another type I cadherin, E-cadherin (CDH1) is expressed on oocyte membranes (*Mora et al., 2012*; *Wang et al., 2009*). β-Catenin is also a transcription factor which plays an important role during follicle development and granulosa cell specification in the ovary (*Nusse and Clevers, 2017*). Constitutive activation of β-catenin in Sertoli cells of the mouse testis drives their trans-differentiation toward ovarian granulosa cell lineage (*Li et al., 2017*), while in ovaries constitutive β-catenin induces elevated expression of the estrogen synthesising gene *Cyp19a1* (*Fan et al., 2010*). YAP1 also plays a critical role in mediating regulation of follicle development through mechanical cues (*Sun and Diaz, 2019*). In granulosa cells YAP1 is required for proliferation and promotes differentiation (*Ji et al., 2017*). However, the respective roles of N- and E-cadherin regulation of β-catenin and YAP1 pathways during ovarian folliculogenesis or oocyte maturation and ovulation has not been investigated.

Ovulation is initiated by a surge of luteinising hormone from the pituitary, inducing dramatic remodelling of the ovarian follicle ECM, and differentiation of cells. The intercellular junctions between cumulus cells and the oocyte are disrupted as the cumulus–oocyte complex (COC) undergoes dynamic expansion of a unique viscoelastic ECM through gene induction in granulosa and cumulus cells (*Robker et al., 2018*; *Bianco et al., 2019*). This COC matrix expansion is essential for ovulation and fertilisation-potential of oocytes (*Robker et al., 2018*; *Russell and Robker, 2007*). The expanded COC acquires an adherent and migratory cell phenotype and can be shown to robustly adhere to ECM components such as fibronectin and collagens that are abundant in the follicle wall (*Akison et al., 2012*).

The localisation of N-cadherin on granulosa cells was recently shown to be involved in primordial follicle formation (reviewed in *Hernandez Gifford, 2015*; *Piprek et al., 2020*), and N-cadherin

antagonists have also been shown to cause apoptosis of cultured granulosa cells (*Makrigiannakis et al., 1999*). We now report previously undiscovered roles for N-cadherin in granulosa cell responses to hormones, as well as COC expansion, oocyte meiotic maturation, and ovulation. Two recently developed, small molecule N-cadherin antagonists (*Mrozik et al., 2020*; *Smits et al., 2020*) were shown to disrupt the interaction between TZPs and oocytes, and to dysregulate β-catenin and YAP1 signalling pathways in mouse COCs, blocking oocyte meiotic maturation. Furthermore, treatment of mice in vivo with an N-cadherin antagonist blocked ovulation with pervasive effects on β-catenin and YAP1 pathways leading to altered response to ovulation stimulus. Our findings demonstrate a role for N-cadherin mechanosensing to maintain ovarian tissue architecture, and hence the hormone responsiveness essential for oocyte maturation and ovulation.

## Results

### N-cadherin antagonists block acquired periovulatory COC adhesion

Through screening a small molecule library, we identified two compounds that block the adhesion of periovulatory COCs to fibronectin using the xCELLigence system. Two potent COC adhesion blocking compounds were: (*S*)-1-(3,4-dichlorophenoxy)-3-(4-((*S*)–2-hydroxy-3-(4-methoxyphenoxy) propyl-amino) piperidin-1-yl)propan-2-ol (CRS-066) (*Smits et al., 2020*) and 5 [3,4 dichlorobenzyl sulfanyl]4H 1,2,4 triazol 3 amine (LCRF-0006) (*Mrozik et al., 2020*), both of which are small molecule N-cadherin antagonists. Dose-dependent inhibition of COC adhesion to fibronectin, compared to vehicle controls, was confirmed at doses above 1 μM for CRS-066 ($IC_{50}$ = 0.4 μM) and above 36 μM for LCRF-0006 ($IC_{50}$ = 38 μM) (*Figure 1a*). Analogues of both compounds were synthesised with substitutions of two key active chloride side chains on the aromatic ring structure, which abolished the COC adhesion blocking activity at the same concentrations (*Figure 1a*). We confirmed the N-cadherin blocking action of the active compounds by showing they reduced N-cadherin abundance at intercellular contacts in N-cadherin expressing SKOV-3 ovarian cancer cell line (*Tang et al., 2016*). Again, significant effects of CRS-066 and LCRF-0006 were observed with 1 and 36 μM, respectively (*Figure 1b–e*). Additionally, N-cadherin-dependent spheroid formation in 67NR mouse breast cancer cells expressing exogenous N-cadherin was blocked by the N-cadherin antagonists (*Figure 1f, g*), while there was no effect on N-cadherin negative cells (*Figure 1—figure supplement 1*).

Together, these data reveal that N-cadherin plays a hitherto unappreciated role in periovulatory COCs and that these new N-cadherin antagonists can disrupt adhesion of periovulatory COCs to substratum (i.e. fibronectin) as well as blocking cell–cell adhesion in other non-ovarian, N-cadherin expressing cells.

### Adherens complexes between cumulus cells and oocytes are disrupted by N-cadherin inhibition

Expression of *Cdh2*, encoding mouse N-cadherin was equivalently high in granulosa cells and COCs with no significant regulation by hormones inducing follicle growth (44 hr eCG), or ovulation, with the exception of a transient fourfold reduction in *Cdh2* at the time of ovulation (12 hr post hCG) specifically in COCs (*Figure 2a*). Expression of *Ctnnb1* encoding the adherens junction cytoskeletal adaptor protein β-catenin was high in granulosa and COCs and maintained throughout folliculogenesis and ovulation (*Figure 2b*). *Cdh1* encoding mouse E-cadherin was fivefold more abundant in COCs than granulosa cells with a significant fivefold reduction in mRNA expression seen 8 and 12 hr after ovulation was induced by hCG (*Figure 2c*).

N-cadherin and β-catenin proteins were abundant on granulosa and cumulus cell surfaces at intercellular junctions (*Figure 2d* indicated by arrows) from primordial follicle stage and throughout folliculogenesis and ovulation. In contrast, both theca interna and externa had no detectable N-cadherin. N-cadherin, and β-catenin were also abundant along cumulus cell TZPs extending across the zona pellucida and contacting the oocyte membrane (arrowheads, *Figure 2d*). Interestingly, N-cadherin and E-cadherin were shown to co-localise at the oocyte plasma membrane by co-staining in whole-mount COCs (*Figure 2e*).

The assembly of functional adherens complexes at granulosa cell interfaces was confirmed using proximity ligation assay (PLA) which showed that N-cadherin and β-catenin are present in the same protein complexes at points of intercellular contact in cultured granulosa cells (*Figure 2—figure*

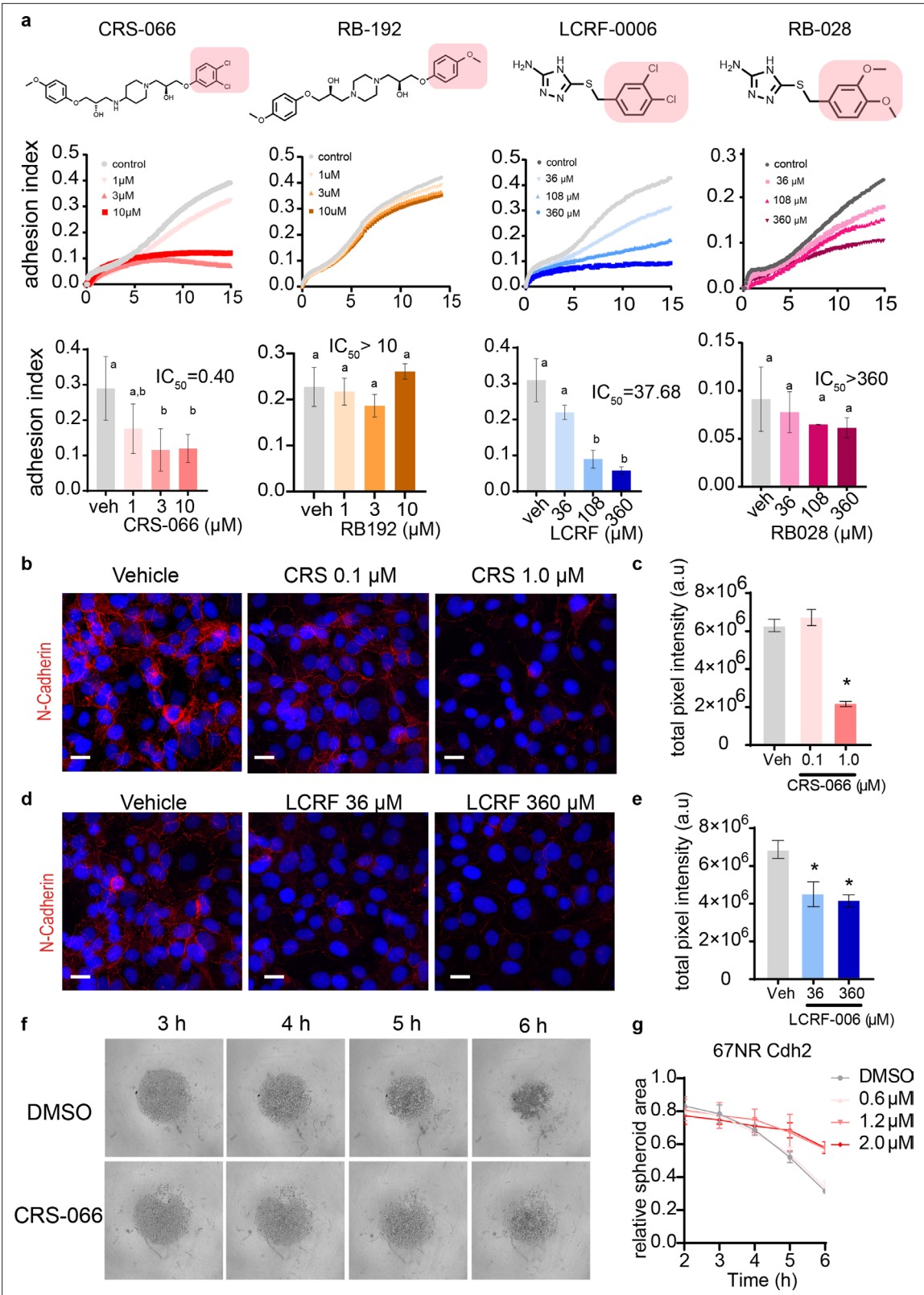

**Figure 1.** N-cadherin controls adhesive capacity of ovulating cumulus–oocyte complexes (COCs). (**a**) Top panel: Chemical structures of N-cadherin antagonists CRS-066, LCRF-0006, and analogues RB-192 and RB-028. Active side chain or modified side chain depicted by highlighted box. Middle panel: Average cell adhesion index values of preovulatory COCs (11 hr post-hCG) interacting with a fibronectin substrate in presence of vehicle or N-cadherin antagonists CRS-066, LCRF-0006, and side chain modified analogues RB-192 and RB-028 or vehicle at respective doses. Cell indices were

*Figure 1 continued on next page*

*Figure 1 continued*

determined using the xCELLigence Impedance system over 15 hr (*n* = 3 independent experiments with 2 technical replicates per treatment). Bottom panel: Mean ± SD adhesion index values at 6 hr different superscript letters indicate significant differences (P<0.05), and $IC_{50}$ values of respective drugs calculated from dose–response curve and compared using one-way ANOVA. (**b, d**) Representative confocal images of N-cadherin adherens junctions on SK-OV-3 cells treated with N-cadherin antagonists CRS-066 (0.1–1.0 μM), LCRF-0006 (36–360 μM), or vehicle for 24 hr (*N* = 3). Scale bar: 40 μM. (**c, e**) Quantification of N-cadherin adherens junctions. Mean of total pixel intensity in red channel, ± SEM of at least 50 cells. *N* = 3 independent experiments. Statistical analyses with two-tailed unpaired *t*-test. * denotes <0.01. (**f**) Bright-field images of spheroid formation in 67NR mouse mammary cell line expressing ectopic Cdh2 were treated with CRS-066 or vehicle at respective time points. Cells were seeded at 2000 cells per well, and formation of spheroids was assessed by imaging every hour for 6 h. (**g**) Mean ± SEM of spheroid area in 67NR-Cdh2 cells treated with either vehicle or increasing doses of CRS-066 (0–2 μM) over 6 hr.

The online version of this article includes the following figure supplement(s) for figure 1:

**Figure supplement 1.** Spheroid formation assay in N-cadherin deficient or N-cadherin expressing mouse 67NR cells.

*supplement 1a*). Likewise, in whole-mount COCs the cumulus cells showed numerous foci of interacting N-cadherin and β-catenin at cell–cell contacts between cumulus cells (*Figure 2—figure supplement 1b*). Notably, we also found PLA signals for N-cadherin and β-catenin interaction at the oocyte plasma membrane indicating adherens complex formation containing N-cadherin and β-catenin at the cumulus cell–oocyte interfaces (*Figure 2—figure supplement 1c*). Since the oocyte contains only E-cadherin, this suggests that heterotypical adherens complexes of N-cadherin and E-cadherin form at the cumulus cell–oocyte interface. Furthermore, we found that cumulus cell N-cadherin is required to stabilise oocyte membrane E-cadherin and β-catenin by incubating COCs with 1 μM CRS-066 or DMSO control for 4 hr. Whole-mount immunostaining showed β-catenin and E-cadherin localisation at the oocyte membrane in controls, while treatment with CRS-066 caused decreased β-catenin on cumulus cell surface, as well as on the oocyte membrane and a concomitant loss of E-cadherin on oocyte membrane (*Figure 2f*).

## N-cadherin antagonists disrupt cumulus expansion and oocyte meiotic maturation

To determine whether the loss of oocyte membrane β-catenin and E-cadherin upon N-cadherin inhibitor treatment disrupts the bi-directional communication between cumulus cells and the oocyte we performed in vitro maturation (IVM) in the presence of the N-cadherin antagonists or vehicle controls and assessed COC expansion and oocyte meiotic maturation.

Increasing concentrations of LCRF-0006 (36–360 μM) dose dependently reduced cumulus expansion index at the end of 12 hr IVM (*Figure 3a, b*). This inhibition of COC expansion after treatment with LCRF-0006 was not due to impaired response to the FSH and EGF stimulus or activation of critical genes. Expression of hyaluronan synthase (*Has2*), which produces the key cumulus matrix component hyaluronan, was in fact dose dependently increased by LCRF-0006 (*Figure 3c*). Expression of known mediators of COC expansion amphiregulin (*Areg*) and the prostaglandin synthase enzyme (*Ptgs2*) (*Takahashi et al., 2006*; *Peluffo et al., 2012*) were either increased or unaffected by LCRF-0006 treatment (*Figure 3c*). mRNA for connective tissue growth factor (*Ctgf*), a canonical YAP1 target, was significantly reduced by the two highest doses of LCRF-0006.

The other N-cadherin antagonist, CRS-066, had a similar effect, leading to a dose-dependent reduction in cumulus expansion during IVM, from 0.1 to 1.0 μM (*Figure 3d, e*). Again, the response to IVM stimulus and induction of critical COC expansion genes was not prevented. The expression of *Areg*, *Ptgs2*, and *Ctgf* were significantly increased by CRS-066 while *Has2* remained unaltered (*Figure 3f*).

RNA-Seq was performed to determine the effect of the N-cadherin antagonists CRS-066 n on global gene expression during maturation of COCs. Principal component analysis showed separate clusters of gene expression in CRS-066 treated and vehicle controls, indicating a consistent effect on cell signalling by the antagonist (*Figure 3—figure supplement 1a*). Differential expression analysis confirmed 168 significant differentially expressed genes (DEGs) (adjusted p-value $<1 \times 10^{-6}$, $\log_2$ fold change ≥±0.5), with 37 genes downregulated and 131 upregulated in CRS-066-treated COCs (*Figure 3—figure supplement 1b*). Gene ontology analyses of the DEGs downregulated by CRS-066 treatment showed that regulation of cell migration and ECM reorganisation were the most overrepresented biological pathways (*Figure 3g*). Protein-lysine 6-oxidase activity, an enzyme responsible

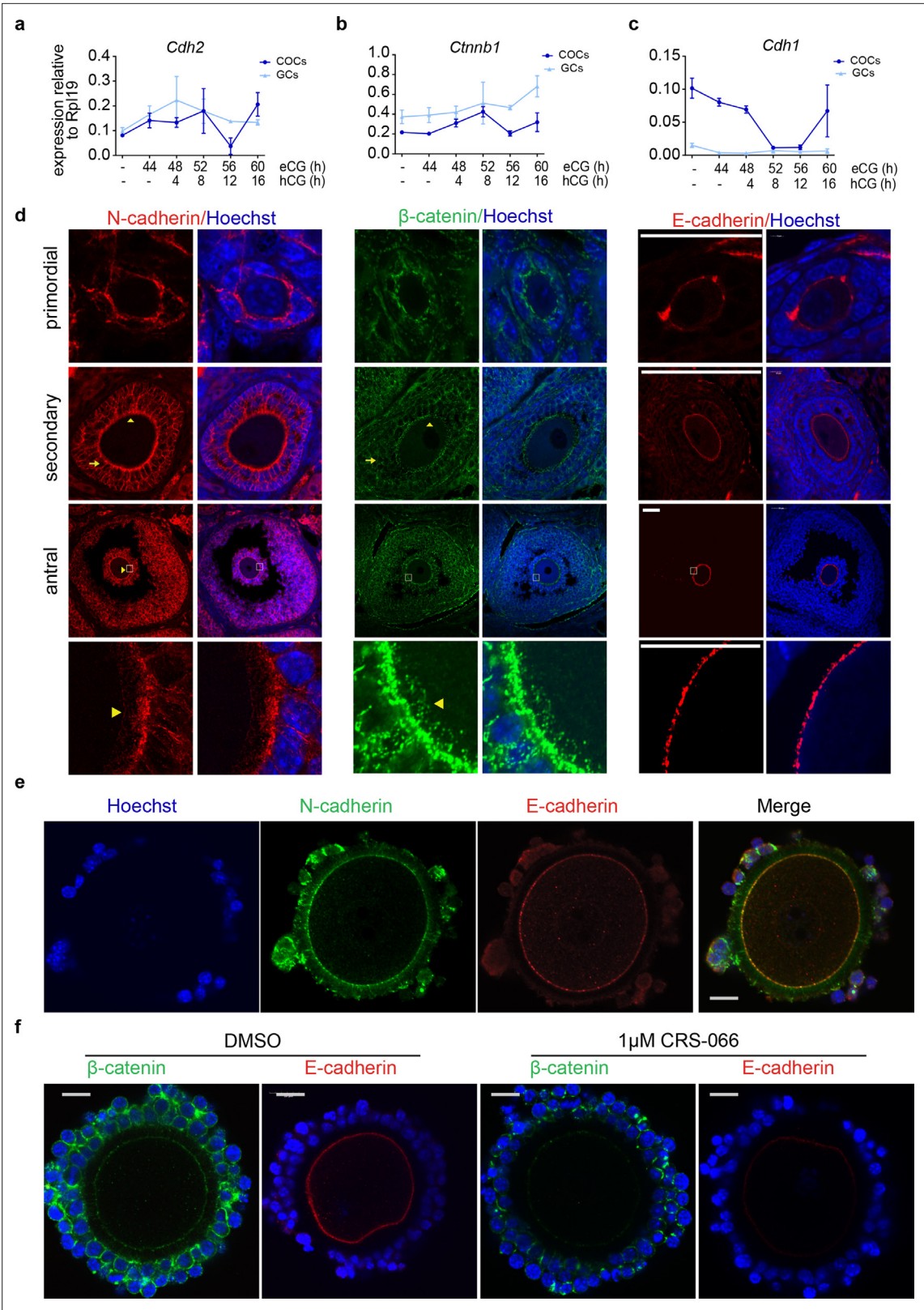

**Figure 2.** N-cadherin maintains oocyte–cumulus cell interaction. (**a–c**) Time-course of Cdh2, ctnnb1, and Cdh1 mRNA expression in isolated granulosa cell (GCs) or cumulus–oocyte complexes (COCs) from mouse ovaries at indicated time points after eCG and hCG stimulation of folliculogenesis and ovulation (*N* = 3 animals per time point). Cdh2 and Ctnnb1 levels are high in GC and COC throughout folliculogenesis, with a transient drop in Cdh2 level 12 hr after ovulation stimulus, while E-cadherin was high in COCs and significantly reduced by ovulation stimulus. The levels shown of the

*Figure 2 continued on next page*

*Figure 2 continued*

indicated mRNAs were determined by TaqMan qPCR normalised to Rpl19. (**d**) Immunofluorescent staining of N-cadherin, β-catenin, and E-cadherin throughout ovarian folliculogenesis. Confocal images of mouse ovarian sections obtained from eCG primed mice and stained using anti N-cadherin (left panel), anti-β-catenin (middle panel), and E-cadherin (right panel). DNA is counterstained with Hoechst. Arrows indicate presence of N-cadherin and β-catenin at granulosa–granulosa cell junctions in secondary and antral follicle stages. Arrowheads indicate presence of N-cadherin and β-catenin at oocyte–cumulus interface. High magnification images show transzonal projections extending from cumulus cells and anchored to oocyte membrane Scale bar: 50 μM. (**e**) Whole-mount immunofluorescent staining showing co-localisation of N-cadherin (green) and E-cadherin (red) at the oocyte plasma membrane in mouse COC from antral follicles of eCG primed mice. N-cadherin is also evident on cumulus cell surfaces and transzonal projections. Cumulus cell and oocyte nuclear DNA are counterstained with Hoechst. Scale bar: 50 μM. (**f**) Whole-mount immunostaining shows loss of β-catenin and E-cadherin at the oocyte plasma membrane after treatment with CRS-066. COCs obtained from antral follicles of eCG primed mice and treated with CRS-066 or vehicle for 4 hr. COCs were fixed and stained with anti-E-cadherin (green) and anti-β-catenin (red). DNA was counterstained with Hoechst. Scale bar: 50 μM.

The online version of this article includes the following figure supplement(s) for figure 2:

**Figure supplement 1.** In situ proximity ligation assay (PLA).

for cross-linking of collagens, was the most over-represented molecular function (*Figure 3h*). Gene set enrichment analysis (GSEA) identified the β-catenin and Hippo signalling pathways canonical downstream signalling pathways of N-cadherin were significantly enriched in CRS-066-treated COC DEGs (*Figure 3i, j*). In agreement with the qPCR analysis, the RNA-Seq results showed treatment with CRS-066 increased the expression of Hippo-pathway responsive genes including *Areg* and *Ctgf* (*Figure 3k*). In contrast, the expression of critical granulosa cell-specific and β-catenin responsive genes *Foxl2* and *Cyp19a1*, were significantly downregulated in COCs with CRS-066 treatment.

To confirm the notion that YAP1 is required for COC maturation we assessed the effect of YAP1 inhibition on cumulus expansion during IVM. Treatment with YAP inhibitor, verteporfin (3 μM), significantly reduced COC expansion with concomitant decrease in the canonical YAP1 target *Ctgf* as well as *Areg*, and *Ptgs2*, while *Has2* expression remained unaffected. These gene expression changes are all consistent with increased YAP1 activity in CRS-066-treated COCs (*Figure 3—figure supplement 1c, d*).

Our findings that N-cadherin is present in intercellular junctions between cumulus cells and oocytes, and that N-cadherin antagonists alter cumulus cell response during IVM, led us to investigate the impact of the antagonists on oocyte meiotic maturation. Both CRS-066 and LCRF-0006 treatment during IVM dose dependently reduced the proportion of oocytes undergoing germinal vesicle breakdown (GVBD) and the proportion reaching MII stage, as determined by the presence of an extruded polar body after 12 hr IVM (*Figure 4*). The stage of meiotic arrest was determined by visualising meiotic spindles and cortical actin in oocytes by labelling β-tubulin, actin staining (phalloidin), and chromosomes (Hoechst). In untreated and vehicle-treated controls >90% of oocytes reached MII stage, as demonstrated by successful assembly of the metaphase II spindle and extrusion of the sister chromatid into polar bodies. Increasing doses of CRS-066 caused more oocyte arrest at GV-intact or MI stage (*Figure 4a–c*). Specifically, 0.3 μM CRS-066 significantly reduced polar body extrusion to 57%, with over 30% oocytes arrested in MI, while 1.0 μM CRS-066 resulted in more than 50% oocytes arresting at GV and the remainder after GVBD. Oocytes exhibiting MI spindle formation were rarely observed. LCRF-0006 treatment also caused oocyte arrest in early meiosis, with 20% arrested in GV stage at the lowest 36 μM dose and almost all oocytes arresting with intact GVs at the highest (360 μM) dose (*Figure 4d–f*).

## N-cadherin antagonist CRS-066 inhibits ovulation in mice

Given the robust effect of CRS-066 and LCRF-0006 on COC adhesion, expansion and meiotic maturation in vitro, we hypothesised that these N-cadherin antagonists may interfere with ovulation in vivo. To assess this, stimulated ovulation was performed in prepubertal (3-week-old) female mice in conjunction with administration of 2× daily intraperitoneally (i.p.) injections of CRS-066 (50 mg/kg), or LCRF-0006 (100 mg/kg) or vehicle for 4 days (*Figure 5—figure supplement 1a*).

CRS-066 treatment significantly compromised the number of oocytes retrieved from oviducts after 16 hCG treatment (11.6 ± 7) compared to vehicle-treated control mice (26.2 ± 6 ovulations/ovary, p < 1 × 10$^{-5}$, n = 6 mice per group) (*Figure 5a*). Histological analysis showed the presence of multiple ovulated luteinised structures in vehicle-treated mice, while CRS-066-treated mice had unruptured

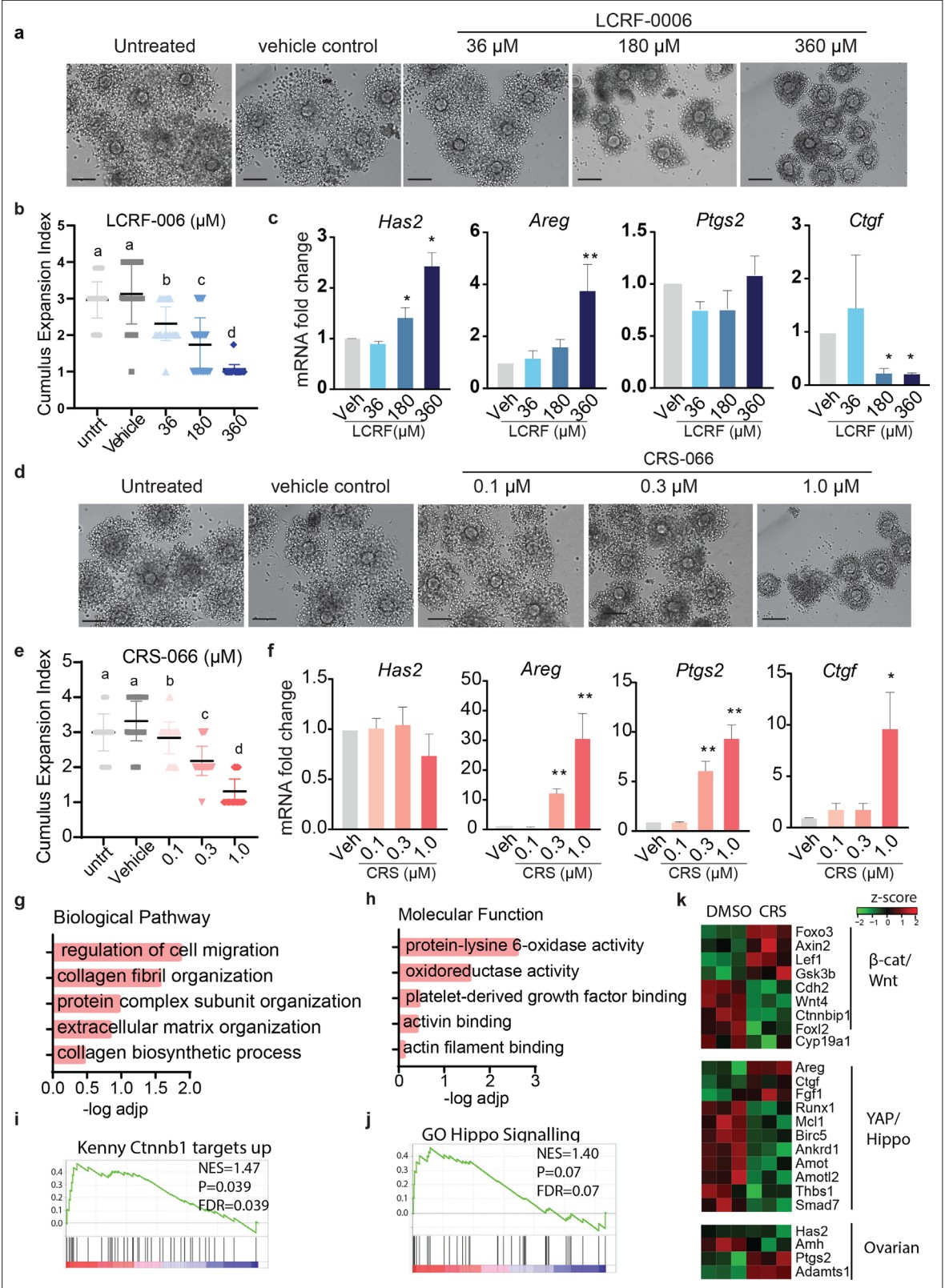

**Figure 3.** N-cadherin antagonists block cumulus expansion in mouse cumulus–oocyte complexes (COCs). COCs from eCG primed mice were treated with LCRF-0006 (36–360 µM) or CRS-066 (0.1–1 µM) during in vitro maturation (IVM) (EGF and FSH stimulated) and cumulus expansion was assessed after 12 hr or gene expression assessed after 10 hr IVM. (**a, d**) Representative bright-field images of COCs after 12 hr IVM treated with LCRF-0006 or CRS-066. Scale bar: 10 µm. (**b, e**) Mean ± SEM of cumulus expansion indices from a and d *n* = >20 COCs per experiment. *N* = 4 independent experiments,

*Figure 3 continued on next page*

*Figure 3 continued*

different superscript letters indicate significant differences (P<0.05), *p < 0.05, **p < 0.01. (**c, f**) Effect of N-cadherin antagonist treatment during IVM (10 hr) on the expression of key genes involved in COC expansion during IVM. Mean ± SEM. $N = 3$ independent experiments. Statistical testing with one-way ANOVA, *p < 0.05, **p < 0.01. (**g, h**) Gene ontology enrichment of biological pathways and molecular functions of significantly differentially downregulated genes identified in RNA-Seq analysis of COCs after CRS-066 (0.3 µM) treatment compared to vehicle treatment. All data are presented as the ratio of CRS-066 over vehicle ($N = 3$). (**I, j**) Gene set enrichment analysis (GSEA) plot demonstrating the upregulation of Ctnnb1 and Hippo signalling pathways in CRS-066- versus vehicle-treated COCs. Net enrichment score (NES) values are shown. $N = 3$ independent biological replicates. (**k**) Heatmap representing the relative expression profiles of transcripts involved in Wnt\β-catenin, Hippo\YAP, and ovarian signalling axes.

The online version of this article includes the following figure supplement(s) for figure 3:

**Figure supplement 1.** N-cadherin antagonist and Yap1 antagonist effect COC expansion and gene expression.

antral follicles with entrapped oocytes, confirming that ovulation was specifically disrupted by CRS-066 treatment (***Figure 5b***). Next, RNA-Seq and differential gene expression analysis in CRS-066- and vehicle-treated whole ovary RNA was performed. Unsupervised hierarchical clustering of RNA-sequencing data identified distinct gene expression profiles in CRS-066 treated compared to control ovaries (***Figure 5c***). A total of 330 significant DEGs were identified with (adjusted p-value $<1 \times 10^{-6}$, $log_2$ fold change $\geq\pm0.5$), with 136 genes upregulated and 194 genes downregulated (***Figure 5—figure supplement 1b, c***). Gene ontology analyses of genes downregulated by CRS-066 treatment showed that ECM organisation and cell–cell adhesion, migration and motility were the most affected pathways (***Figure 5d***). Integrin binding and cadherin binding were the most affected molecular functions with CRS-066 treatment (***Figure 5e***). GSEA showed a profile enriched for both β-catenin target gene set and Hippo signalling pathways (***Figure 5f, g***). Of note, CRS-066 repressed the expression of a group of ovarian genes known to be in the β-catenin/Wnt signalling pathway, including *Cyp19a1* (gene encoding aromatase which is involved in estrogen synthesis) that was repressed >2-fold, while *Foxl2*, a forkhead transcription factor specifically expressed in granulosa cells, and Amh, a granulosa-specific hormone were upregulated >1.5-fold. Other β-catenin/Wnt signalling pathway genes, including Wnt inhibitory factor (*Wif1*), and glycogen synthase kinase (*GSK3b*) which phosphorylates β-catenin were also downregulated after CRS-066 treatment (not shown). CRS-066 treatment also repressed Hippo pathway genes *Areg* and *Ctgf* (***Figure 5h***).

Consistent with transcriptomic analysis, qPCR showed that genes involved in granulosa cell hormone responsiveness including the gonadotropin receptor genes (*Fshr* and *Lhcgr*) were unaffected (***Figure 5i***), while an elevation in *Amh* and *Foxl2* (***Figure 5k***) expression in whole ovary RNA likely indicates elevated granulosa cells numbers in CRS-066-treated ovaries that fail to ovulate. Expression of oocyte-specific genes *Gdf9* and *Nobox* were unaffected by CRS-066 treatment (***Figure 5i***). Among genes required for ovulation, *Areg* and *Ptgs2* expression was significantly reduced, while *Pgr* and *Adamts1* were unaffected (***Figure 5j***).

The numbers of follicles at each developmental stage were equivalent in CRS-066- and vehicle-treated mice. Ovaries from mice treated in the same manner and collected either before hCG treatment (eCG 44 hr) or 11 hr after hCG (1 hr before ovulation) showed equivalent numbers of follicles at each stage of development from primary to antral. Typical periovulatory follicle morphology and number confirmed that the response to hCG was normal (***Figure 5l***). Immunostaining of N-cadherin showed a disruption to the normal organisation of N-cadherin at intercellular boundaries in the granulosa layers, loss of N-cadherin in the TZPs and dissociation of cumulus cells from the cumulus–oocyte interface in follicles of mice treated in vivo with CRS-066 (***Figure 5m***). Proliferation and apoptosis in granulosa cells, assessed by Ki-67 and cleaved caspase 3 (CC3) immunostaining were not significantly different between ovaries of vehicle and CRS-066-treated mice (***Figure 5n***). The effects of CRS-066 treatment on expression of specific genes in ovaries collected 44 hr after eCG confirmed a normal response to hormone treatments with expression of Cyp19a1 high in preovulatory (eCG 44 hr treated) ovaries, and repressed by hCG stimulation of ovulation, while Ptgs2 expression was low at preovulatory stage, but induced by hCG 11 hr. Treatment with CRS-066 at both pre- and periovulatory stages of folliculogenesis caused a significant reduction in Cyp19a1 and Ptgs2, but an increase in Amh expression compared to vehicle controls (***Figure 5o***), consistent with the observations in hCG 16 hr post-ovulatory ovaries.

Treatment of mice with 100 mg/kg LCRF-0006 for 4 days prior to ovulation resulted in similar numbers of ovulated oocytes retrieved from oviducts (23.9 ± 11.2) compared to vehicle controls (25.6

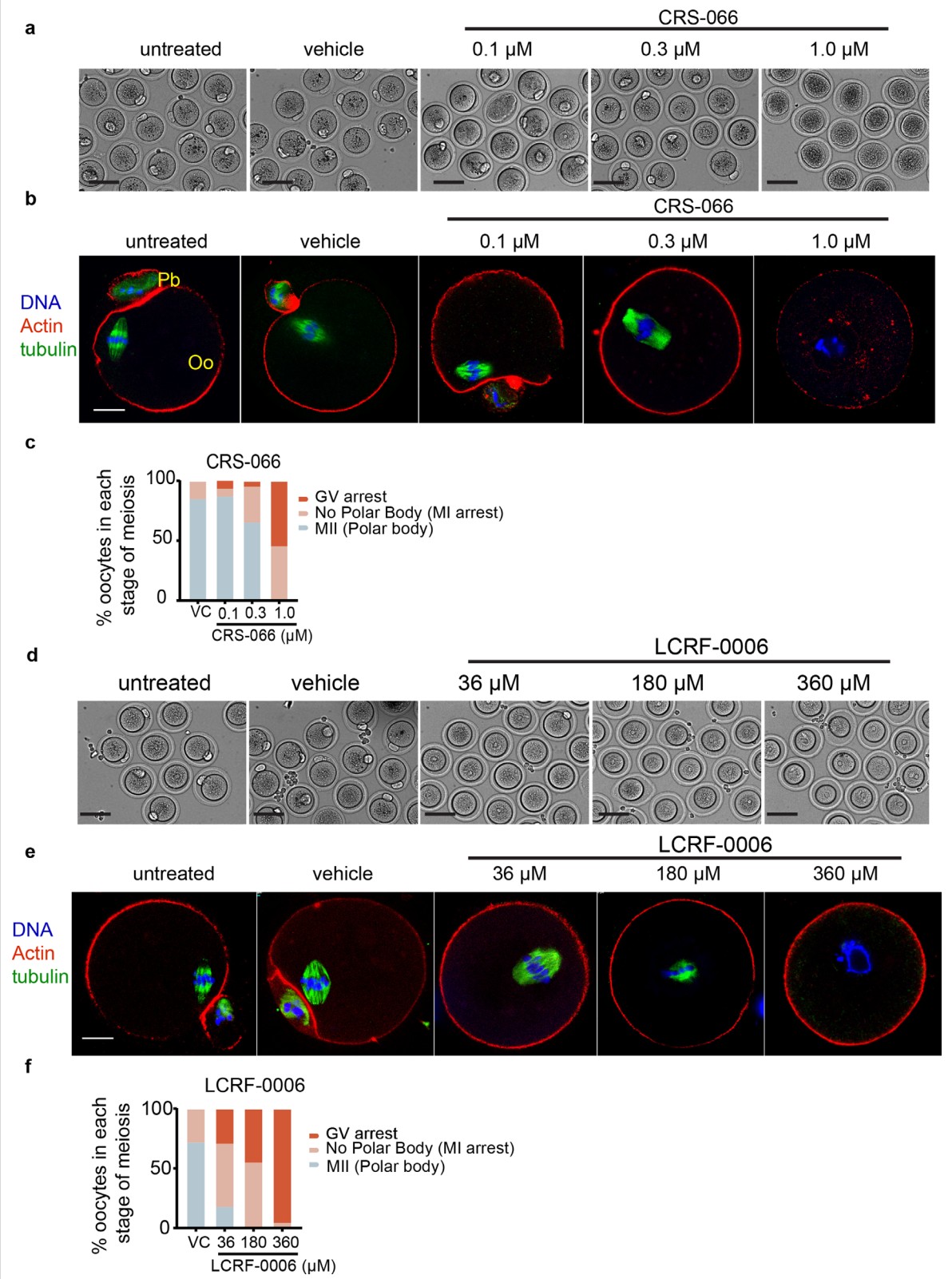

**Figure 4.** N-cadherin antagonists block meiotic maturation in mouse cumulus–oocyte complexes (COCs). COCs from eCG primed mice were treated with LCRF-0006 (36–360 µM) or CRS-066 (0.1–1 µM) during in vitro maturation (IVM) (12 hr EGF and FSH stimulated), then oocytes were denuded and meiotic stage was assessed by labelling actin (phalloidin, red), spindles (tubulin IF, green), and DNA (Hoechst, blue). n = >20 COCs per experiment; N = 4 independent experiments. (**a, d**) Representative bright-field images of denuded mouse COCs showing oocyte and polar body morphology after

*Figure 4 continued on next page*

*Figure 4 continued*

vehicle, CRS-066 or LCRF-0006 treatment. Scale bar: 5 µM. (**b, e**) Representative confocal fluorescent images of denuded mouse oocytes after IVM treated with LCRF-0006 or CRS-066 as indicated, showing polar bodies, spindle structure, or germinal vesicle morphology typical in each treatment condition. Scale bar: 20 µM. Pb indicates polar body formation; Oo indicates oocyte. (**c, f**) Percent of oocytes at each stage of meiotic progression from $n > 20$ oocytes per experiment in 4 independent experiments. MI defined by the any evidence of tubulin staining in the first metaphase spindle with no polar body extrusion; MII defined by presence of a polar body and MII metaphase plate. GV defined by oocyte DNA lacking any tubulin staining to indicate MI spindle formation.

± 5.3 ovulations per ovary, p = 0.67, *n* = 6 mice per group) after 16 hr hCG treatment. Additionally, there were no significant changes in *Cyp19a1*, *Ptgs2*, or *Areg* gene expression, indicating this compound was less active in vivo as was also the case in vitro.

## Granulosa-specific N-cadherin knockout inhibits ovulation in mice

To verify that N-cadherin is the specific target of the antagonists that modulate mechanosensitive gene expression and ovulation, we generated granulosa-specific Cdh2 knockout. Breeding crosses of *Cdh2^Fl/+^;Amhr2^Cre/+^* mice producing offspring with the expected genotypes at Mendelian ratios. Female offspring with *Cdh^Fl/+^;Amhr2^Cre^*, versus *Cdh^Fl/FL^;Amhr2^Cre^* (granulosa-specific N-cadherin depletion) were comparatively assessed in preovulatory (eCG 44 hr) ovaries. qPCR analysis confirmed significant fourfold reduction in Cdh2 mRNA expression in *Cdh^Fl/FL^;Amhr2^Cre^* ovaries (*Figure 6a*). Likewise, immunofluorescence of ovarian sections showed dramatically reduced N-cadherin protein in ~80–90% of granulosa cells in all follicles. A mosaic pattern was observed with 10–20% of granulosa cells in antral follicles showing approximately normal levels of N-cadherin immunostaining (*Figure 6b–e*), consistent with other reports of incomplete penetrance of the Amhr2-cre transgene expression in granulosa cells (*Kyrönlahti et al., 2011*; *Hernandez Gifford et al., 2009*). The N-cadherin depleted areas of follicles showed multiple regions of detached cells suggesting a loss of intercellular adhesion. The expression of putative N-cadherin mechanoresponsive genes Areg and Ptgs2 were significantly (p < 0.05) reduced 2- to 3-fold, respectively, in the granulosa-specific Cdh2 mutants, and Cyp19a1 showed trending (p = 0.08, 2.5-fold) reduced expression (*Figure 6a*), similar to the reduced Cyp19a1 expression reported in *Ctnnb1^Fl/FL^;Amhr2^Cre^* mice (*Hernandez Gifford et al., 2009*). Ovulation examined by collecting and counting COCs in oviducts of female *Cdh^Fl/FL^;Amhr2^Cre^* mice after eCG + hCG 14 hr stimulation was significantly (p < 0.01, threefold) reduced (*Figure 6f*) and histological assessment identified multiple large unruptured follicles in the *Cdh^Fl/FL^;Amhr2^Cre^* granulosa-specific Cdh2 mutant ovaries (*Figure 6g*). Together these results support the conclusion that N-cadherin is important for cell–cell adhesion and mecahnosignalling within granulosa cell populations, and for successful ovulation of the mature preovulatory follicles.

## Discussion

Our study shows, for the first time, that N-cadherin plays a central role during late follicular morphogenesis. Specifically, N-cadherin antagonists, or Cdh2 gene mutation disrupt intercellular contacts between granulosa cells, as well as oocyte–cumulus cell contacts leading to impaired COC expansion, oocyte maturation, and ovulation. Likewise, even incomplete disruption of the Cdh2 gene expression led to a similar disruption to mechanoresponsive gene expression and ovulation. Broad transcriptional changes indicate cross-talk between N-cadherin with β-catenin and Hippo/YAP pathways mediating a mechanical signal that integrates with the hormone actions to regulate genes required for oocyte maturation and ovulation.

There is evidence that the ovarian follicle is a mechanosensitive organ. Either excess or inadequate tissue rigidity profoundly affects folliculogenesis and oocyte maturation. This is evident in Polycystic Ovarian Syndrome (PCOS), a prevalent infertile condition where excess rigid ECM prevents follicle and oocyte growth progression (*Monniaux et al., 2020*). It is also shown in vitro, where isolated follicles in different bioengineered support matrices exhibit reduced follicle growth and viability in soft or rigid environments (*West et al., 2007*). Our findings suggest that N-cadherin is a key mechanosensor and transducer of growth and differentiation of granulosa cells, cumulus cells, and oocytes. Consistent with this, dysregulation of β-catenin and Hippo/Yap1 has been shown in conditions of disrupted follicle growth including PCOS (*Ji et al., 2017*), as well as Primary Ovarian Insufficiency (POI)

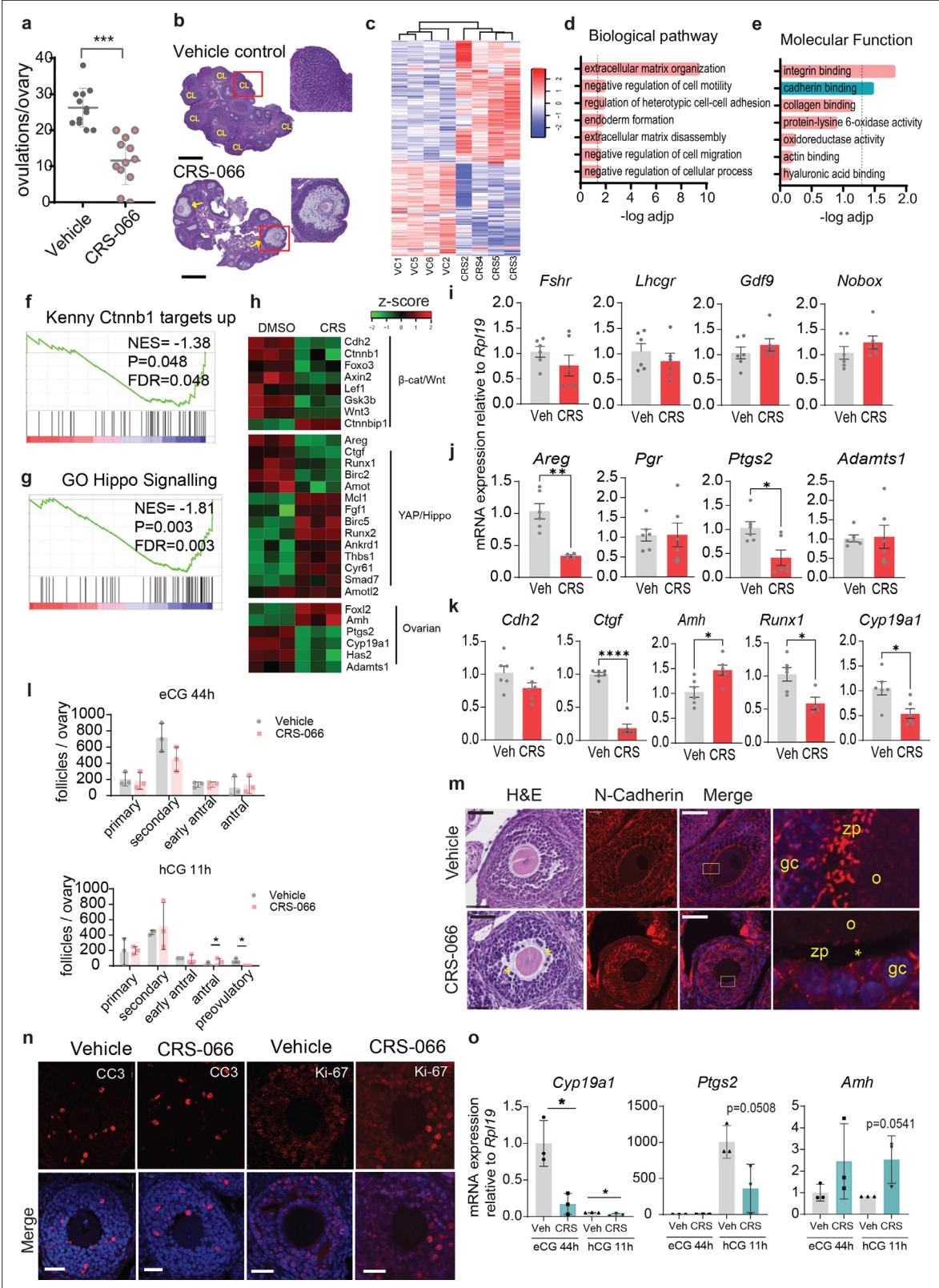

**Figure 5.** N-cadherin antagonist CRS-066 blocks ovulation in vivo. (**a**) Ovulation rate of 21-day-old mice treated with CRS-066 (50 mg/kg) or vehicle (7.5% DMSO in 0.9% saline). Cumulus–oocyte complexes (COCs) in oviducts counted 16 hr after hCG injection. Graph represents mean ± SEM from *N* = 6 animals; ***p < 0001 (unpaired two-tailed *t*-test). (**b**) Histology of ovaries by haematoxylin and eosin staining. CL indicates corpus leuteum; arrows indicate trapped oocytes in CL. Scale bar: 100 μm. (**c**) Hierarchical clustering of RNA-sequencing analysis results shows differentially expressed genes

*Figure 5 continued on next page*

*Figure 5 continued*

between CRS-066- and vehicle-treated mice (*N* = 6 mice per treatment). Gene ontology enrichment of biological pathways (**d**) and molecular functions (**e**) of significantly differently downregulated genes in CRS-066-treated mouse ovaries compared to vehicle treated ovaries. (**f, g**) Gene set enrichment analyses (GSEA) plot shows downregulation of Ctnnb1 and Hippo signalling pathways in CRS-066-treated mice ovaries compared to vehicle-treated ovaries. Net enrichment score (NES) values are shown. (**h**) Heatmap representing the relative expression profiles of transcripts involved in Wnt\β-catenin, Hippo\YAP and ovarian signalling axes in CRS-066-treated ovaries compared to vehicle. *N* = 3 biological replicates. Relative mRNA expression of key genes involved in gonadotrophin signalling and oocyte function (**i**), COC expansion and ovulation (**j**), or folliculogenesis (**k**) hr in ovaries treated with CRS-066 compared to vehicle treatment and determined by quantitative reverse transcription PCR (qRT-PCR). Bar graph show mean ± SEM. *N* = 6 ovaries from independent CRS or vehicle-treated mice. Statistical testing with Student's *t*-test; *p < 0.05; **p < 0.01; ****p < 0.00001. (**l**) Follicle counts at primary, secondary, pre-antral, antral, and ovulatory stages in ovaries from mice treated with either CRS-006 (50 mg/kg) or vehicle control (7.5% DMSO). *N* = 3 mice/treatment/time point. (**m**) Representative follicle morphology H&E (left) section and N-cadherin immunofluorescence (right) section in mice treated with either CRS-066 or vehicle. H&E and immunofluorescence highlight disorganised granulosa cells organisation. Asterisks indicate loss of transzonal projections between oocyte and cumulus cells. Scale bar: 30 µm. (**n**) Representative confocal immunofluorescent images of mouse ovaries stained with anti-cleaved caspase 3 and anti-Ki-67. Scale bar: 30 µm. (**o**) Relative mRNA expression of key genes involved in oocyte growth and ovulation in CRS-066- or vehicle-treated mice (*N* = 3/treatment) at either 44 hr post eCG or 11 hr post hCG.

The online version of this article includes the following figure supplement(s) for figure 5:

**Figure supplement 1.** N-cadherin antagonist treatment and gene expression changes in vivo.

and fibrosis (***Kawamura et al., 2013***; ***Matsuzaki and Darcha, 2013***). Further studies are required to determine the interplay between cadherins and integrins in the cells of ovarian follicles, as both cell surface receptor groups interact with the actin-based microfilaments and regulate similar biological processes.

Direct physical contacts between cumulus cells and the oocyte are important for transport of cAMP and cGMP via gap junctions to maintain oocyte meiotic arrest. Both in vivo and in vitro N-cadherin antagonists caused disruption of TZPs and cumulus–oocyte cellular contacts in COCs. Disruption of these contacts was recently shown to be a normal event in response to induced COC maturation (***Abbassi et al., 2021***). The effects of N-cadherin antagonists provide the first evidence confirming that N-cadherin stabilises these unique cell–cell contacts and that premature disruption of the cumulus cell–oocyte contacts is highly detrimental to follicle growth and oocyte maturation. Our observation that two N-cadherin antagonists severely impaired COC expansion and meiotic maturation in IVM while not depleting expression of cumulus matrix genes such as *Has2* suggests that N-cadherin mediates signalling between oocyte and cumulus cells, transducing essential signals necessary for these preovulatory events. This is in line with N-cadherin being a known mechanotransducer and being abundant at cell junctions between cumulus cells and in TZPs at cumulus–oocyte contacts. The disruption of cumulus cell arrangement around oocytes in ovaries of N-cadherin antagonist treated mice also demonstrates the necessity for N-cadherin to stabilise cell–cell contact and maintain the tissue architecture critical for bidirectional cell communication. The possibility of heterotypic interaction of N- and E-cadherin at the cumulus cell–oocyte interface suggests a unique type of intercellular junction in the context of oocytes and cumulus cells with interactomes that are also likely to be specialised. The molecular composition of these adherens complexes, and the mechanism behind cadherin-driven intracellular signalling in the COC clearly warrants further investigation.

β-Catenin transduces Wnt signalling, and in ovarian cells it is fundamental to specification of the female somatic cell lineage (***Li et al., 2017***) establishing female sex determination and follicle formation (reviewed in ***Hernandez Gifford, 2015***). Our studies are the first to demonstrate that N-cadherin is important for functional gene regulation in the follicle by regulating the β-catenin and Hippo/Yap1 pathways. Reduced expression of β-catenin regulated granulosa cell-specific genes, including *Ptgs2* and *Cyp19a1* (***Fan et al., 2010***; ***Hernandez Gifford et al., 2009***; ***Bai et al., 2017***; ***Boyer et al., 2010***), and changes in the transcriptome profile indicate the loss of β-catenin activity caused by the N-cadherin antagonist. This is consistent with N-cadherin's reported role stabilising β-catenin at tensile cell junctions and facilitating nuclear translocation in response to Wnt signals (***Nelson and Nusse, 2004***; ***Fan et al., 2018***). Thus, our results demonstrate that tissue structure and mechanical force contribute to hormone responsiveness and the steroidogenic phenotype of granulosa cells. We did not see evidence of upregulated testis specific genes indicating that neither N-cadherin antagonist treatment, nor gene depletion was not sufficient to reverse granulosa cell specification as seen with complete β-catenin ablation (***Chassot et al., 2008***). However, our results do indicate that N-cadherin stabilises granulosa and cumulus cell phenotypes through β-catenin signalling.

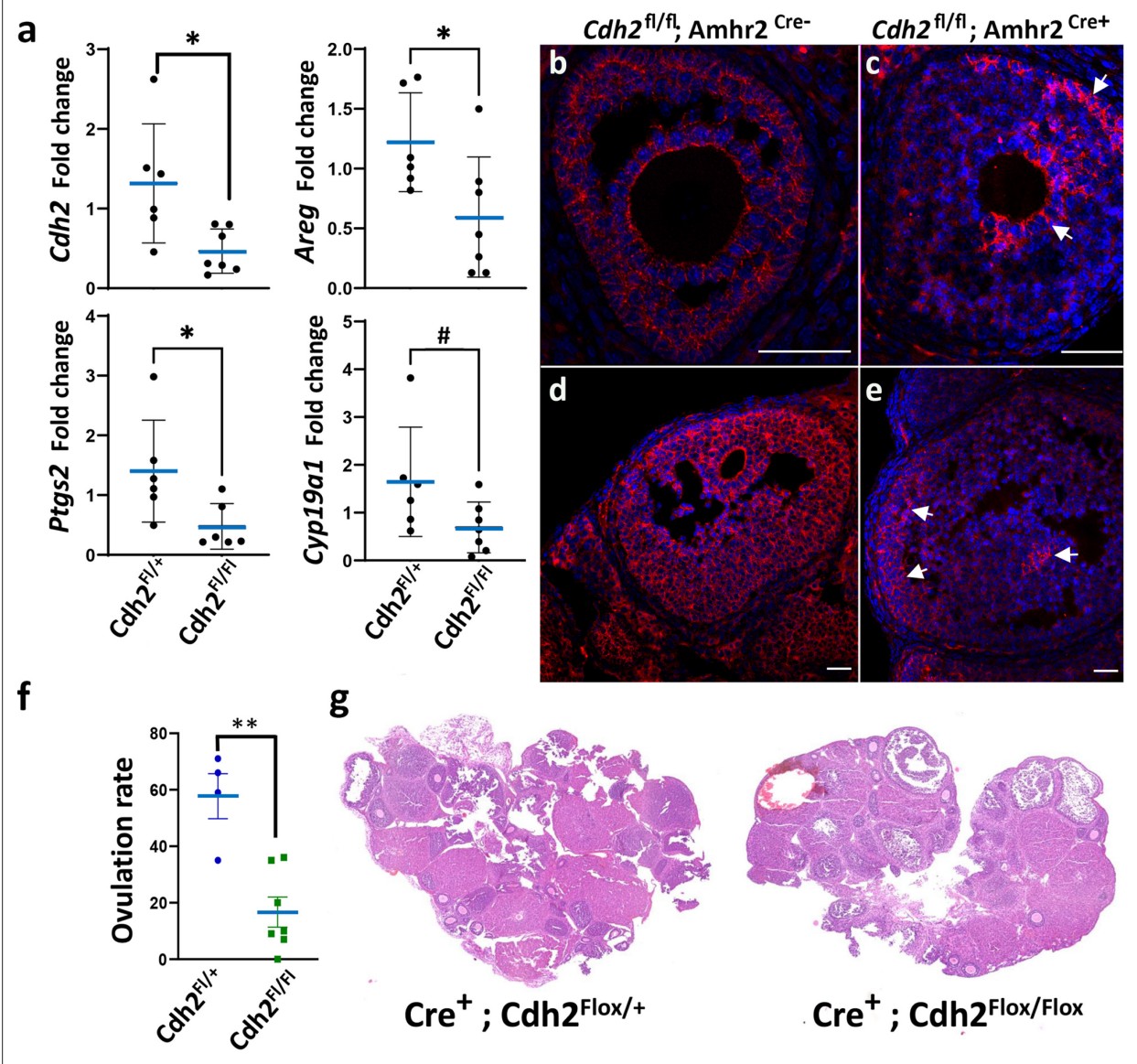

**Figure 6.** Granulosa conditional Cdh2 null mutation disrupts ovarian gene expression and blocks ovulation. (**a**) qPCR analysis of relative mRNA expression of Cdh2, Areg, Ptgs2, and Cyp19a1 in ovaries of control (*Cdh2^Fl/+; Amhr2^Cre*) and granulosa-specific Cdh2 null mutants (*Cdh2^Fl/Fl; Amhr2^Cre*), n = 6 individual animals, *p < 0.05, #p < 0.08. (**b–e**) Immunofluorescent analysis of N-cadherin protein in ovaries of control (*Cdh2^Fl/+; Amhr2^Cre*) and granulosa-specific Cdh2 null mutants (*Cdh2^Fl/Fl; Amhr2^Cre*), showing mosaic depletion of N-cadherin in granulosa cells of mutant follicles. Arrows indicate mosaic regions with persistent N-cadherin. (**f**) Ovulation rate of 21-day-old mice with indicated control or granulosa-specific mutant genotypes. Cumulus–oocyte complexes (COCs) in oviducts counted 16 hr after hCG injection. Graph represents mean ± SEM from N = 4 and 7 animals, respectively; **p < 0.01 (unpaired two-tailed *t*-test). (**g**) Histology of ovaries by haematoxylin and eosin staining. Scale bar: 100 µm.

N-cadherin antagonists also reduced expression of Yap1 target genes with key functions in folliculogenesis (*Ctgf* and *Areg*) (*Fan et al., 2018*), indicating a loss of Yap1 function in vivo. These findings suggest that an ovarian Yap1 mechanosensing pathway, also mediated through N-cadherin, is attenuated in the antagonist treated mice. Trans-cellular interaction of cadherins has been shown to sequester Yap1 in the cytoplasm (*Gumbiner and Kim, 2014*; *Hirate et al., 2013*; *Kim et al., 2011*), and peptide inhibitors of N-cadherin can block nuclear translocation of Yap1 in response to tissue rigidity in mesenchymal stem cells (*Cosgrove et al., 2016*). Thus, our results show Yap1 signalling in granulosa cells is controlled by N-cadherin and regulates the differentiation status of these cells and their response to hormone stimuli. This conclusion is also supported by the morphological change in follicles of Wnt4-deficient mice (*Prunskaite-Hyyryläinen et al., 2014*), and in granulosa-specific Yap1

knockout mice (*Lv et al., 2019*). Together, these findings suggest that mechanotransduction signals through N-cadherin mediate the reported high Yap1 activity in granulosa cells needed to sustain normal folliculogenesis.

During IVM of COCs, N-cadherin antagonists blocked COC expansion, and oocyte meiosis. CRS-066 paradoxically increased *Areg* and *Ptgs2*, which are β-catenin repressed genes in COCs (*Fan et al., 2010*). Their induction is consistent with compromised β-catenin function after N-cadherin inhibition in vitro as was seen in vivo. The induction of *Areg*, *Ptgs2*, and the canonical Yap1 target, *Ctgf* all suggest activation of Yap1 by N-cadherin antagonist in IVM. Supporting this, the Yap1 antagonist verteporfin caused dose-dependent repression of *Ctgf*, *Areg*, and *Ptgs2* indicating Yap1 indeed mediates induction of these critical COC maturation genes. This is consistent with reports that show that close contact with oocytes suppresses Yap1 activation in cumulus cells, and that cumulus expansion elicits a mechanical signal that releases this inhibition leading to expression COC maturation genes (*Sun and Diaz, 2019*). Thus, N-cadherin antagonist-mediated inhibition of mechanotransduction disrupted the normal regulation of Yap1. Ctgf mediates size regulation of many organs and is a known ovarian mitogen that directly regulates follicle development and ovulation by upregulating activin-dependent LOX-mediated extracellular remodelling (*Nagashima et al., 2011*). N-cadherin antagonism during IVM and in vivo consistently showed that Lysyl oxidase, activin binding and ECM remodelling were among the most affected molecular functions, further supporting our proposed model of N-cadherin acting through Hippo/Yap1/Ctgf to regulate COC expansion, meiotic maturation, and ovulation. The inverse effect of LCRF-0006 on *Ctgf* expression in IVM suggests the two antagonists differ not only in their potency but have different effects on N-cadherin signalling responses. Further investigation is required to determine whether this observation is explained by the targeting of different binding pockets of N-cadherin by LCRF-0006 and CRS-066. Since Yap1 and β-catenin have divergent effects on *Ptgs2* and *Areg and* are both influenced by N-cadherin antagonists, the activation/inactivation state of these individual pathways is difficult to unravel, however it is clear the net effect mediated through N-cadherin cell contacts plays a key role in mediating follicle growth and differentiation.

In conclusion, our study identified N-cadherin as a mechanosensory regulator important in ovarian granulosa cell differentiation and response to hormone stimuli both in vivo and in vitro. The action of N-cadherin through β-catenin and Yap1 signal transducers places new emphasis on the role of these pathways as mediators of hormone responses that also incorporates mechanical cues such as the size and physical tension within the ovary, which is influenced by the neighbouring growing follicles as well as ECM composition. Dysregulation of this mechanism has important implications for conditions such as premature ovarian failure, PCOS, and fibrosis as well as other forms of unexplained infertility. Additionally, the potential to target this signalling pathway with small molecule inhibitors, to block ovulation is an illustration the principle of non-hormonal ovulation blocking contraceptives.

## Methods
### Reagents and antibodies
Unless otherwise stated, reagents were purchased from Sigma-Aldrich (St. Louis, MO, USA).

### Study approval
All animal procedures were approved by the University of Adelaide Animal Ethics Committee (Ethics approval IDs M-2018-071 and M-2018-117) and conducted in accordance with the Australian Code of Practice for the Care and Use of Animals for Scientific Purposes.

### Animals and drug treatments
Hybrid F1 (CBA female/C57 male) prepubertal female mice (21-day-old, weighting 10–12 g) were purchased from Laboratory Animal Services (University of Adelaide) and were maintained on a 12-hr light and 12-hr dark cycle, with rodent chow and water provided ad libitum. Mice were injected i.p. with equine chorionic gonadotropin at 5 IU/0.1 ml saline (0.9%) per 12 g of bodyweight, followed 44 hr later by i.p. injection of human chorionic gonadotropin at 5 IU/0.1 ml saline per 12 g of body-weight. The N-cadherin antagonists CRS-066 and LCRF-0006 were a kind gift from Zonula Inc, Kirkland, Quebec, Canada. CRS-066 (compound-15 di-succinate salt) was injected i.p. at 50 mg/kg in 0.05 ml saline and LCRF-0006 was injected i.p. at 100 mg/kg of in 0.05 ml saline containing 40%

2-hydroxyproprol-β-cyclodextrin twice daily for four consecutive days starting the day prior to treatment with eCG.

## Conditional granulosa Cdh2-null mice

Cdh2 floxed mice (B6.129S6(SJL)-*Cdh2*[tm1Glr/J]) with exon 1, containing the translational start site of the Cdh2 gene flanked by two loxP sites (*Kostetskii et al., 2005*; *Jamin et al., 2002*) were obtained from Jackson Laboratories. Mice were genotyped by performing PCR using the following primers: Cdh2F 5′-CCAAAGCTGAGTGTGACTTG-3′ and Cdh2R 5′-TACAAGTTTGGGTGACAAGC-3′. To generate animals with Cdh2 gene ablation Cdh2 floxed mice were crossed with Amhr2-Cre knock in mice B6;129S7-*Amhr2*[tm3(cre)Bhr] (*Jamin et al., 2002*) genotyped by PCR using the following primers: Amhr2-Cre-F 5′-CCTGGAAATGCTTCTGTCCG-3′ and Amhr2-Cre R 5′-CAGGGTGTTATAAGCAATCCC-3′.

## Cell line and growth conditions

The human ovarian cancer cell lines SK-OV-3 and 67NR were cultured in Dulbecco's modified Eagle's media supplemented with 10% (vol/vol) FCS, 2 mM L-glutamine, 100 U ml⁻¹ penicillin, and 100 μg ml⁻¹ streptomycin at 37°C in an atmosphere of 5% $CO_2$ in air.

67NR cells were transduced with C-terminal EGFP-tagged murine N-cadherin lentiviral construct. The N-cadherin-EGFP construct was cloned into the pCDH-CMV-CMV-MCS-EF1a-Puro lentivirus vector. Cells were transduced with $5 \times 10^4$ IU/ml and selected by flow cytometry using EGFP fluorescence to isolate N-cadherin expressing cells.

## xCELLigence real-time cell adhesion assay

Assay of the induced COC adhesion to fibronectin was performed using the xCELLigence system (ACEA Biosciences) in E-plate 16 according to the manufacturer's instruction as previously described with little modification (*Akison et al., 2012*). Briefly, The E16 xCELLigence plates were prepared by coating with 50 μl of mouse recombinant fibronectin (5 μg/ml) diluted in αMEM medium before incubation for a minimum of 1 hr at 37°C/5% $CO_2$. Wells were washed twice with 100 μl/well of αMEM medium. Drugs were added to each well at required concentrations in 50 μl bi-carbonate-buffered αMEM/1% FCS medium. Plates were inserted into the xCELLigence station, and the base-line impedance was measured. Freshly isolated COCs from CBA-F1 mice at 11 hr post-hCG, just prior to ovulation, were seeded into each well in 50 μl of bicarb-buffered αMEM and electircal impedance was measured every 5 min until 15 hr after seeding. Each treatment was run in duplicate and each plate contained untreated COC control wells. Results are presented as adhesion index over time as well as the concentration eliciting 50% inhibition of adhesion response for $N = 3$ independent biological replicates.

## Spheroid formation assay

Spheroid assay was conducted in 96-well U bottomed ultra-low adhesion Nunclon Sphera plates (Thermo Scientific). Sub-confluent N-cadherin-EGFP 67NR cells were dissociated with TrypLE, washed with warm PBS and counted with the Countess cell counter (Invitrogen). Cells were seeded at 2000 cells per well and centrifuged at $300 \times g$ for 3 min before being placed in culture and imaged every 60 min from 2 to 6 hr after seeding on the Incucyte live cell imager (Essen Bioscience) using the spheroid assay module. Spheroid area was calculated using ImageJ plugin INSIDIA (*Moriconi et al., 2017*).

## Proximity ligation assay

PLA was performed on cultured granulosa cells or intact COCs using the Duolink PLA Probes and PLA Fluorescence in situ Detection Kit Red as per the manufacturer's protocol. Briefly, cultured granulosa cells or COCs were fixed in 4% paraformaldehyde (PFA) and permeabilised with PBS + 0.01% Triton X-100 for 1 hr at room temperature. Cells were blocked with blocking Buffer for 1 hr at 37°C and incubated with primary antibody couple (N-cadherin + β-catenin) diluted in Antibody Diluent for 2 hr at room temperature or overnight at 4°C. Then, cells were incubated with PLA probes of appropriate species for 1 hr at 37°C, oligo probes were ligated for 30 min at 37°C and the amplification reaction was carried out at 37°C for a minimum of 100 min. Between steps, cells were washed using the provided wash buffers. Slides were mounted with Prolong Gold Mounting Media, cured for at

least 1 hr in the dark and were stored at –20°C prior to imaging. Slide imaging was done through the Olympus confocal microscope FV1000.

## Isolation of ovaries, COCs, and granulosa cells

For the time-course of gene regulation, mice were untreated or were hormonally stimulated by intra-peritoneal injection with 5 IU equine chorionic gonadotropin (eCG) followed by 5 IU hCG at 46-hr post eCG. Mice were culled at the following time points: unstimulated (no eCG and hCG), 0 hr (no hCG), 4, 8, 10, and 12 post hCG. Ovaries were dissected and granulosa cells were collected by repeated puncturing of the ovaries.

Ovaries or oviducts were dissected from mice and placed in HEPES-buffered minimum essential medium alpha (αMEM) supplemented with 1% (vol/vol) foetal calf serum at 37°C. Ovulated COCs were isolated from the oviducts of mice at 16 hr after hCG injection as indicated, placed in HEPES-buffered α-MEM supplemented with 1% (vol/vol) foetal calf serum and counted under a stereo micro-scope. After counting, ovulated COCs were snap-frozen for RT-qPCR analyses.

One ovary per mouse was fixed in 4% PFA (wt/vol) in PBS [80 mM $Na_2HPO_4$, 20 mM $NaH_2PO_4$, and 100 mM NaCl (pH 7.5)] for 24 hr and processed into paraffin blocks that was then sectioned (5 μm) and stained with haematoxylin and eosin. Images were captured at high resolution using NanoZoomer Digital Pathology technology (Hamamatsu Photonics K.K.). The other ovary was snap frozen in liquid nitrogen and stored at –80°C for RNA analysis.

## Gene expression analyses

Total RNA was isolated using Trizol (Thermo Fisher Scientific), as per the manufacturer's instructions, with the inclusion of 15 μg GlycoBlue (Ambion) during precipitation. Total RNA was then treated with 1 U of DNase as per the manufacturer's instructions. RNA concentration and purity were quantified using a Nanodrop ND-1000 Spectrophotometer (Biolab Ltd, Victoria, Australia). First-strand comple-mentary DNA was synthesised from total RNA using random hexamer primers (Geneworks) and Super-script III reverse transcriptase (Invitrogen). Quantitative reverse transcription PCR was performed using Taqman gene expression assays (Applied Biosystems) and reactions were run in duplicate on an AB7900HT Fast PCR System using manufacturer's recommended amplification settings. Gene expres-sion levels were normalised to a reference gene (Rpl19).

## Immunofluorescence

Immunohistochemistry and immunofluorescence were performed on 4% PFA-fixed paraffin-embedded 5 μm sections mounted onto Super frost microscope slides (Thermo Fisher Scientific) or whole mounts of denuded oocytes. Tissue sections were dewaxed in xylene and rehydrated. Antigen retrieval was performed by incubating section in either citrate buffer (10 mM sodium citrate, pH 6.0) or in Tris-EDTA buffer (10 mM Tris/1 mM EDTA/0.05% (vol/vo) Tween 20, pH 9.0) for 20 min at 95°C. After cooling, sections were washed with TBS containing 0.025% Tween-20 (TBST, pH 7.6) for 10 min and blcoked in 10% normal goat serum (NGS) in TBST for 1 hr at RT. Sections were probed with primary antibodies against N-cadherin (BD Biosciences; Cat# 610920; 1:500); β-catenin (CST; Cat# 8480; 1:500), E-cad-herin (CST, Cat# 14472; 1:500); a-tubulin (Thermo Fisher; Cat# 236-10501); Ki-67 (CST; Cat# 9449; 1:1000); and Cleaved caspase 3 (CST; Cat# 9661; 1:1000) diluted in 10% NGS and incubated O/N at at 4°C overnight in a humid chamber. Negative controls were performed using matching isotype control either rabbit IgG (CST; Cat# 3900S) or mouse IgG (CST; Cat# 5415S) at equivalent concentration used for primary antibodies. Sections were washed three times for 5 min each in TBST and incubated with secondary antibodies Alexa fluor 647, Alexa Fluor 488, and Alexa Fluor 594 (Thermo Fisher Scientific; Cat# A21244, A11034, A32728, A110001; 1:2000) for 1 hr at RT alongside Hoeschst 33342 (Thermo Fisher Scientific; Cat# H3570; 1:250). Sections were washed three times for 5 min each in TBST and mouted with fluorescence mounting medium (Dako, Santa Clara, USA). Images of immunofluores-cence secitons were captured by confocal microcopy FV1000 (Olympus, Tokyo, Japan).

## RNA sequencing and data processing

Total RNA was extracted from COCs and whole ovaries by homogenising in Trizol as described above. mRNA was converted to strand specific Illumina compatible sequencing libraries using the Nugen Universal Plus mRNA mRNA-Seq library kit (Tecan, Mannedorf, Switzerland) as per the manufacturer's

instructions. Libraries were sequenced on the Illumina Nextseq550 High output mode and v2.5 chemistry platform for 75 bp single-end reads. Sequencing quality control was performed using FastQC. Sequences were mapped to the GENCODE mouse transcriptome (GRCm38.p6, M25) and mapped transcripts were quantified using Salmon. Clustering analysis was performed on count data using variance stabilising transformation as part of the DESeq2 package. Read count was transformed to log counts per million for visualisation via heatmap using edgeR package (R/Bioconductor). Differential expression was analysed using DESeq2. Genes that had a highly stringent FDR $\leq 10^{-6}$ and log fold change $\geq \pm 0.5$ were determined to be DEGs. Biological significance of DEGs was explored by GO term enrichment analysis including biological process and moleuclar function, based on Enrichr web application. GSEA of the genes differentially expressed upon CRS-066 treatment was done using a pre-ranked list on the Gene Set Enrichment Analysis software (software.broadinstitute.org/gsea/index.jsp). Gene sets were extracted from the Molecular Signatures database (The Molecular Signatures Database, v7.3) using the search term 'β-catenin', 'Hippo'. FDR $q$ value was used to rank the results. Gene sets enriched at FDR $q$ value $\leq 0.05$ and nominal p < 0.05 were considered statistically significant. The raw data of all sequencing libraries generated in this study have been submitted to the Gene Expression Omnibus that can be accessed with the number of GSE168347 (ovary) and GSE168348 (COCs).

## Statistical analysis

Statistical analysis was performed using GraphPad Prism 7.0 software (GraphPad Software Inc, La Jolla, CA, USA). Differences between two groups were calculated using the unpaired two-tailed Student's *t*-test. Statistical analyses of three or more groups were compared using one-way ANOVA followed by Bonferroni's multiple comparisons test. Reported values are the mean ± standard error of the mean of three or more independent biological experiments or as indicated. *p < 0.05 and **p < 0.01.

## Acknowledgements

The authors acknowledge the instruments and technical assistance of Microscopy Australia at Adelaide Microscopy, The University of Adelaide. This work was supported by grants from the Bill and Melinda Gates Foundation Contraceptive Discovery Program (OPP1171844, INV-001616). DLR is supported by NHMRC Senior Research Fellowship APP1110562. RLR is supported by NHMRC Senior Research Fellowship APP1117976. KM is supported by an Early Career Cancer Research Fellowship from Cancer Council SA's Beat Cancer Project on behalf of its donors and the State Government of South Australia through the Department of Health and Wellbeing.

## Additional information

### Competing interests

Orest W Blaschuk: Holds shares in Zonula Inc. Darryl L Russell: Guest Editor eLife. The other authors declare that no competing interests exist.

### Funding

| Funder | Grant reference number | Author |
| --- | --- | --- |
| Bill and Melinda Gates Foundation | OPP1171844 | Darryl L Russell |
| Bill and Melinda Gates Foundation | INV-001616 | Darryl L Russell |
| National Health and Medical Research Council | Senior Research Fellowship APP1110562 | Darryl L Russell |
| National Health and Medical Research Council | Senior Research Fellowship APP1117976 | Rebecca L Robker |

| Funder | Grant reference number | Author |
|---|---|---|
| Cancer Council South Australia | Beat Cancer Project Early Career Cancer Research Fellowship | Krzysztof M Mrozik |

The funders had no role in study design, data collection, and interpretation, or the decision to submit the work for publication.

## Author contributions

Alaknanda Emery, Investigation, Methodology, Writing – review and editing; Orest W Blaschuk, Resources, Writing – review and editing; Doan T Dinh, Data curation, Formal analysis, Writing – review and editing; Tim McPhee, Formal analysis, Investigation, Methodology; Rouven Becker, Resources, Formal analysis, Investigation, Methodology, Writing – review and editing; Andrew D Abell, Supervision, Project administration, Writing – review and editing; Krzysztof M Mrozik, Resources, Methodology; Andrew CW Zannettino, Resources, Supervision; Rebecca L Robker, Resources, Supervision, Funding acquisition, Project administration, Writing – review and editing; Darryl L Russell, Conceptualization, Formal analysis, Supervision, Funding acquisition, Investigation, Methodology, Writing – original draft, Project administration, Writing – review and editing

## Author ORCIDs

Orest W Blaschuk ![ORCID] https://orcid.org/0000-0002-9448-4979
Darryl L Russell ![ORCID] https://orcid.org/0000-0002-4930-7658

## Ethics

All animal procedures were approved by the University of Adelaide Animal Ethics Committee (Ethics approval IDs M-2018-071 and M-2018-117) and conducted in accordance with the Australian Code of Practice for the Care and Use of Animals for Scientific Purposes.

Reviewer #1 (Public Review): https://doi.org/10.7554/eLife.92068.2.sa1
Reviewer #2 (Public Review): https://doi.org/10.7554/eLife.92068.2.sa2
Author response https://doi.org/10.7554/eLife.92068.2.sa3

# Additional files

## Supplementary files

MDAR checklist

## Data availability

All data generated or analysed during this study are included in the manuscript and supporting files. Sequencing data have been deposited in GEO under accession codes GSE168347 and GSE168348.

The following datasets were generated:

| Author(s) | Year | Dataset title | Dataset URL | Database and Identifier |
|---|---|---|---|---|
| Emery A, Dinh TD, Robker RL, Russell DL | 2024 | Gene expression analyses of COCs treated with N-Cadherin antagonist CRS-006 | https://www.ncbi.nlm.nih.gov/geo/query/acc.cgi?acc=GSE168347 | NCBI Gene Expression Omnibus, GSE168347 |
| Emery A, Dinh TD, Robker RL, Russell DL | 2024 | Gene expression analyses of whole ovaries from mice that were treated with N-Cadherin antagonist CRS-006 | https://www.ncbi.nlm.nih.gov/geo/query/acc.cgi?acc=GSE168348 | NCBI Gene Expression Omnibus, GSE168348 |

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
