## [Editor Report · eLife assessment]

The manuscript describes **important** findings regarding the significance of CHD2 in ovarian folliculogenesis. Overall, the results lead to **convincing** conclusions, with minimal concerns raised by the reviewers. Both the results and conclusions are well discussed. This work will be of interest to ovarian biologists and physicians working on female fertility.

---

## [Referee Report · Reviewer #1 (Public Review)]

Summary:

This manuscript reports experiments designed to dissect the function of N-cadherin during mammalian folliculogenesis, using the mouse as a model system. Prior studies have shown that this is the principal cadherin expressed by the follicular granulosa cells. Two main strategies are used - small-molecule inhibitors that target N-cadherin and a conditional knockout where the gene encoding N-cad is deleted in granulosa cells. The authors also take advantage of the ability to reproduce key events of folliculogenesis, such as oocyte meiotic maturation, in vitro. Four main conclusions are drawn from the studies: (i) cadherin-based cell contact is required to maintain cadherin (N-cad in the granulosa cells; E-cad in the oocyte) at the plasma membrane; (ii) N-cad is required for cumulus layer expansion; (iii) N-cad is required for meiotic maturation of the oocyte; (iv) N-cad is required for ovulation.

Strengths:

The experiments are logically conceived, clearly described and presented, and carefully interpreted. A key strength of the paper is that multiple approaches have been used (drugs, knockouts, immunofluorescence, PLA, in vitro and in vivo studies). Taken together, they clearly establish essential roles for N-cadherin during folliculogenesis.

It is intriguing that, when cadherin activity is impaired, the cadherins are lost from the plasma membrane. This suggests that, in a multicellular context, interactions with other cadherins, either in cis within the same cell or in trans with a neighboring cell, are required to maintain cadherins at the membrane. Hence, beyond their significance for understanding female reproductive biology, these experiments have broader implications for cell biology.

Weaknesses:

A few points could be considered or clarified by the authors:

The YAP experiments were confusing to the reviewer. CRS-066 increased YAP activity, as indicated by increased expression of target genes. Since CRS-066 prevents expansion, this result suggests that YAP antagonizes expansion. Therefore, blocking YAP should favor expansion. Yet, the YAP inhibitor impaired expansion. In the reviewer's eyes, these results seem to be contradictory.

It is intriguing that the inhibitors were able to efficiently block oocyte maturation. Oocytes from which the cumulus granulosa cells have been removed (denuded) will mature in vitro in the absence of LH or EGF. Since the effect of the inhibitors is to break the contact between the cumulus cells and oocyte, one might have predicted that this would not impair the ability of the oocytes to mature. Perhaps the authors could comment on this.

Regarding the experiments where the inhibitors were administered intra-peritoneally, the authors might comment on the rationale for choosing the doses that were used. An additional point to consider is that, since N-cadherin is expressed in a variety of tissues, an effect of interfering with N-cadherin at these non-ovarian sites could indirectly influence ovarian function.

---

## [Referee Report · Reviewer #2 (Public Review)]

Summary:

The manuscript entitled "N-cadherin mechanosensing in ovarian follicles controls oocyte maturation and ovulation" aimed to investigate the role of N-cadherin in different ovarian physiological processes, including cumulus oocyte expansion, oocyte maturation, and ovulation. The authors performed several in vitro and in vivo mice experiments, using diverse techniques to reinforce their results.

First, they identified two compounds (N-cadherin antagonists) that block the adhesion of periovulatory COCs to fibronectin through screening a small molecule library, using the xCELLigenceTM system, performing proper and complementary controls. Second, the authors showed the presence of N-cadherin adherens junctions between granulosa cells and cumulus cells and at the interface of cumulus cell transzonal projections and the oocyte throughout folliculogenesis. And that these adherens complexes between cumulus cells and oocytes were disrupted when inhibited N-cadherin, as observed by nice representative confocal images. Then, the authors assessed COC expansion and oocyte meiotic maturation to determine whether the loss of oocyte membrane β-catenin and E-cadherin upon N-cadherin inhibitor treatment disrupts the bi-directional communication between cumulus cells and the oocyte. Indeed, N-cadherin antagonists disrupted both processes (cumulus expansion and oocyte meiotic). However, the expression of known mediators of COC expansion (E.g., Areg and Ptgs2) were either increased or unaffected. Nevertheless, RNA-Seq showed consistent effects on cell signaling mRNA genes by the antagonist CRS-066.

In vivo studies using mice were also achieved using stimulated protocols (together with one of the antagonists or vehicle) or granulosa-specific Cdh2 Knockouts to further analyze the role of N-cadherin. N-cadherin antagonist CRS-066 (but not LCRF-0006) significantly reduced mouse ovulation compared to controls. RNA-sequencing data analysis identified distinct gene expression profiles in CRS-066 treated compared to control ovaries. Ovulation in CdhFl/FL; Amhr2Cre mice after stimulation were also significantly reduced; multiple large unruptured follicles were observed in these granulosa-specific Cdh2 mutant ovaries, and the mRNA expression of Areg and Ptgs2 were reduced.

The authors conclude that their study identified N-cadherin as a mechanosensory regulator important in ovarian granulosa cell differentiation able to respond to hormone stimuli both in vivo and in vitro, demonstrating a critical role for N-cadherin in ovarian follicular development and ovulation. They highlighted the potential to inhibit ovulation by targeting this signaling mechanism.

Strengths:

This remarkable manuscript is very well designed, performed, and discussed. The authors analyzed different aspects, and their data supports their conclusions.

Weaknesses:

This study was performed using the mouse as a research model; further studies in larger animals and humans would be interesting and warranted.

Minor comments:

Some results are intriguing. While the AREG y PTGS2 mRNA increased within the COC in vitro by the N-cadherin antagonists, in vivo, the treatment induced a significant increase in both genes when analyzing the whole ovary. What are the authors' ideas that could explain these discrepancies in outcomes?

The authors stated that the ovaries from mice treated in the same manner and collected either before hCG treatment (eCG 44 h) or 11 h after hCG showed equivalent numbers of follicles at each stage of development from primary to antral. However, in Panel l from Figure 5, there is a significant increase in the number of antral follicles in the CRS-066 group (hCG 11 h) compared to the vehicle. Could the author discuss it in the manuscript?

---

## [Author Response]

**Public Reviews:**

**Reviewer #1 (Public Review):**
Summary:This manuscript reports experiments designed to dissect the function of N-cadherin during mammalian folliculogenesis, using the mouse as a model system. Prior studies have shown that this is the principal cadherin expressed by the follicular granulosa cells. Two main strategies are used - small-molecule inhibitors that target N-cadherin and a conditional knockout where the gene encoding N-cad is deleted in granulosa cells. The authors also take advantage of the ability to reproduce key events of folliculogenesis, such as oocyte meiotic maturation, in vitro. Four main conclusions are drawn from the studies: (i) cadherin-based cell contact is required to maintain cadherin (N-cad in the granulosa cells; E-cad in the oocyte) at the plasma membrane; (ii) N-cad is required for cumulus layer expansion; (iii) N-cad is required for meiotic maturation of the oocyte; (iv) N-cad is required for ovulation.Strengths:The experiments are logically conceived, clearly described and presented, and carefully interpreted. A key strength of the paper is that multiple approaches have been used (drugs, knockouts, immunofluorescence, PLA, in vitro and in vivo studies). Taken together, they clearly establish essential roles for N-cadherin during folliculogenesis.It is intriguing that, when cadherin activity is impaired, the cadherins are lost from the plasma membrane. This suggests that, in a multicellular context, interactions with other cadherins, either in cis within the same cell or in trans with a neighboring cell, are required to maintain cadherins at the membrane. Hence, beyond their significance for understanding female reproductive biology, these experiments have broader implications for cell biology.Weaknesses:A few points could be considered or clarified by the authors:The YAP experiments were confusing to the reviewer. CRS-066 increased YAP activity, as indicated by increased expression of target genes. Since CRS-066 prevents expansion, this result suggests that YAP antagonizes expansion. Therefore, blocking YAP should favor expansion. Yet, the YAP inhibitor impaired expansion. In the reviewer's eyes, these results seem to be contradictory.

The mechanism through which N-cadherin inhibitors block cumulus expansion isn’t fully elucidated but isn’t deemed to be through YAP alone. The transcriptional changes indicate crosstalk between N-cadherin, β-catenin and Hippo/YAP pathways, as well as impacting on the signalling between cumulus cells and the oocyte.

It is intriguing that the inhibitors were able to efficiently block oocyte maturation. Oocytes from which the cumulus granulosa cells have been removed (denuded) will mature in vitro in the absence of LH or EGF. Since the effect of the inhibitors is to break the contact between the cumulus cells and oocyte, one might have predicted that this would not impair the ability of the oocytes to mature. Perhaps the authors could comment on this.

Indeed, removal of cumulus cells permits oocyte meiotic maturation by reducing oocyte cAMP, leading to activation of meiosis promoting factor (MPF). A hypothesis would be that cyclic nucleotides and MPF arrest in the oocyte are maintained when N-cadherin contacts are blocked but this was not determined.

Regarding the experiments where the inhibitors were administered intra-peritoneally, the authors might comment on the rationale for choosing the doses that were used. An additional point to consider is that, since N-cadherin is expressed in a variety of tissues, an effect of interfering with N-cadherin at these non-ovarian sites could indirectly influence ovarian function.

Doses were chosen based on previous reported use of these inhibitors in vivo (Mrozik et al. 2020). Possible effects of the N-cadherin antagonists in other tissues was a carefully considered in this and the previous Mrozik et al study. While we saw no evidence of effects in gross morphological observations, or closer examination of vasculature or blood in these studies, this potential is not excluded.

**Reviewer #2 (Public Review):**
Summary:The manuscript entitled "N-cadherin mechanosensing in ovarian follicles controls oocyte maturation and ovulation" aimed to investigate the role of N-cadherin in different ovarian physiological processes, including cumulus oocyte expansion, oocyte maturation, and ovulation. The authors performed several in vitro and in vivo mice experiments, using diverse techniques to reinforce their results.First, they identified two compounds (N-cadherin antagonists) that block the adhesion of periovulatory COCs to fibronectin through screening a small molecule library, using the xCELLigenceTM system, performing proper and complementary controls. Second, the authors showed the presence of N-cadherin adherens junctions between granulosa cells and cumulus cells and at the interface of cumulus cell transzonal projections and the oocyte throughout folliculogenesis. And that these adherens complexes between cumulus cells and oocytes were disrupted when inhibited N-cadherin, as observed by nice representative confocal images. Then, the authors assessed COC expansion and oocyte meiotic maturation to determine whether the loss of oocyte membrane β-catenin and E-cadherin upon N-cadherin inhibitor treatment disrupts the bi-directional communication between cumulus cells and the oocyte. Indeed, N-cadherin antagonists disrupted both processes (cumulus expansion and oocyte meiotic). However, the expression of known mediators of COC expansion (E.g., Areg and Ptgs2) were either increased or unaffected. Nevertheless, RNA-Seq showed consistent effects on cell signaling mRNA genes by the antagonist CRS-066.In vivo studies using mice were also achieved using stimulated protocols (together with one of the antagonists or vehicle) or granulosa-specific Cdh2 Knockouts to further analyze the role of N-cadherin. N-cadherin antagonist CRS-066 (but not LCRF-0006) significantly reduced mouse ovulation compared to controls. RNA-sequencing data analysis identified distinct gene expression profiles in CRS-066 treated compared to control ovaries. Ovulation in CdhFl/FL; Amhr2Cre mice after stimulation were also significantly reduced; multiple large unruptured follicles were observed in these granulosa-specific Cdh2 mutant ovaries, and the mRNA expression of Areg and Ptgs2 were reduced.The authors conclude that their study identified N-cadherin as a mechanosensory regulator important in ovarian granulosa cell differentiation able to respond to hormone stimuli both in vivo and in vitro, demonstrating a critical role for N-cadherin in ovarian follicular development and ovulation. They highlighted the potential to inhibit ovulation by targeting this signaling mechanism.Strengths:This remarkable manuscript is very well designed, performed, and discussed. The authors analyzed different aspects, and their data supports their conclusions.Weaknesses:This study was performed using the mouse as a research model; further studies in larger animals and humans would be interesting and warranted.

Indeed, this would be interesting. Ongoing research into therapeutic applications of N-cadherin targeting is reviewed in Blaschuk OW. Front Cell Dev Biol. 2022 Mar 3;10:866200

Minor comments:Some results are intriguing. While the AREG y PTGS2 mRNA increased within the COC in vitro by the N-cadherin antagonists, in vivo, the treatment induced a significant increase in both genes when analyzing the whole ovary. What are the authors' ideas that could explain these discrepancies in outcomes?

Comparing the responses in IVM COCs to in vivo whole ovaries carries multiple caveats, though as noted, the observations are consistent with altered mechanotransduction in each case. It is important to note the change in pre-ovulatory follicle gene expression in vivo, which likely affects the response of follicles to ovulatory stimulus.

The authors stated that the ovaries from mice treated in the same manner and collected either before hCG treatment (eCG 44 h) or 11 h after hCG showed equivalent numbers of follicles at each stage of development from primary to antral. However, in Panel l from Figure 5, there is a significant increase in the number of antral follicles in the CRS-066 group (hCG 11 h) compared to the vehicle. Could the author discuss it in the manuscript?

A small change in these follicle types was significant in hCG 11h treated mice and is consistent with the altered response to the ovulatory stimulus and reduced ovulation resulting in persistent antral follicles.

**Recommendations For The Authors:**

**Reviewer #1 (Recommendations For The Authors):**
Is the mechanism by which the small molecules block N-cad's adhesive activity known? And is the stable residence of cadherins in the plasma membrane known to depend on their engagement with other cadherins either in cis or in trans?

Adhesion interactions between N-cadherin in Cis or Trans results in their clustering and enrichment at the membrane. Molecular docking models of the small molecule N-cadherin inhibitors are not available. However, these inhibitors were designed as peptidomimetics of the N-cadherin amino-terminus that is shown to interacts in Trans with N-cadherin on neighbouring cells (Blaschuk OW. Front Cell Dev Biol. 2022 Mar 3;10:866200).

Since the inhibitors are blocking cadherin activity, one might have expected the cumulus cell mass to disaggregate into individual cells. Yet, Figures 3a and 3c show that this does not happen. Could the authors speculate how the cells are being held together?